# Decision and navigation in mouse parietal cortex

Michael Krumin[1], Julie J Lee[1], Kenneth D Harris[2], Matteo Carandini[1]*

[1]UCL Institute of Ophthalmology, University College London, London, United Kingdom; [2]UCL Institute of Neurology, University College London, London, United Kingdom

**Abstract** Posterior parietal cortex (PPC) has been implicated in navigation, in the control of movement, and in visually-guided decisions. To relate these views, we measured activity in PPC while mice performed a virtual navigation task driven by visual decisions. PPC neurons were selective for specific combinations of the animal's spatial position and heading angle. This selectivity closely predicted both the activity of individual PPC neurons, and the arrangement of their collective firing patterns in choice-selective sequences. These sequences reflected PPC encoding of the animal's navigation trajectory. Using decision as a predictor instead of heading yielded worse fits, and using it in addition to heading only slightly improved the fits. Alternative models based on visual or motor variables were inferior. We conclude that when mice use vision to choose their trajectories, a large fraction of parietal cortex activity can be predicted from simple attributes such as spatial position and heading.

DOI: https://doi.org/10.7554/eLife.42583.001

## Introduction

Posterior parietal cortex (PPC) is recognized as a key nexus of sensorimotor integration (*Milner and Goodale, 2006*), and the large literature concerning its functions has led to disparate views as to its role. These views originate from a broad variety of experiments, some performed in primates and others in rodents.

A number of studies have related PPC activity to the control of body movement (*Andersen and Buneo, 2002*; *Andersen and Cui, 2009*; *Andersen and Mountcastle, 1983*; *Bisley and Goldberg, 2010*; *Cohen and Andersen, 2002*; *Park et al., 2014*). For instance, neurons in monkey PPC can be influenced by movements of eyes, head, limbs, and body (*Cohen and Andersen, 2002*), by the intention to execute such movements (*Andersen and Buneo, 2002*), or by the attention devoted to the resulting position (*Bisley and Goldberg, 2010*). Neurons in rat PPC, moreover, can encode attributes of body posture (*Mimica et al., 2018*).

Another set of experiments indicates a role of PPC in decision making, especially for decisions based on vision (*Andersen and Cui, 2009*; *Erlich et al., 2015*; *Goard et al., 2016*; *Gold and Shadlen, 2007*; *Katz et al., 2016*; *Latimer et al., 2015*; *Licata et al., 2017*; *Platt and Glimcher, 1999*; *Raposo et al., 2014*; *Sugrue et al., 2004*). Studies in rodents found decision signals to be widespread in PPC populations, where they are mixed with other signals (*Goard et al., 2016*; *Pho et al., 2018*; *Raposo et al., 2014*). One study, in particular, probed memory-based decision signals and found them to be remarkably common, with each PPC neuron firing only for a particular decision and only at a particular moment in a stereotyped sequence (*Harvey et al., 2012*).

A third set of studies, performed in rodents, suggested an important role of PPC in spatial navigation (*McNaughton et al., 1994*; *Nitz, 2006*; *Nitz, 2012*; *Save and Poucet, 2000*; *Save and Poucet, 2009*; *Whitlock et al., 2012*; *Wilber et al., 2014*). Neurons in rat PPC encode combinations of spatial location and body movement (*McNaughton et al., 1994*; *Nitz, 2006*; *Nitz, 2012*;

**\*For correspondence:**
m.carandini@ucl.ac.uk

**Competing interests:** The authors declare that no competing interests exist.

**eLife digest** When we step out of our homes in the morning, we scan our surroundings to decide which path we should take. It is still unclear whether we use different brain areas to examine the environment, decide on a route, and then set our trajectory, or if a single region can play a role in all three processes. An area in the top of our brain, named the posterior parietal cortex (PPC), may be an intriguing candidate: some studies find that this region is involved in vision, others highlight that it takes part in decision-making, and a third group of experiments shows that it is important for setting paths. Could the PPC be integrating all three types of information, or might the activity of the neurons in this area be better explained by just one of these processes?

Here, Krumin et al. trained mice to use visual clues to navigate a virtual reality maze, where they have to 'walk' down a corridor and then choose to 'go down' the right or the left arm. The rodents move on a ball suspended in mid-air, which acts as a treadmill. Meanwhile, the head of the animal is kept still, making it easier to image the activity of hundreds of neurons in the PPC.

The experiments show that when the mouse uses its vision to choose its trajectory, the PPC does not encode visual signals or abstract decisions. Instead, two navigational attributes – the position of the animal along the corridor and its heading angle – activate neurons in this area. Knowing these two features was enough for Krumin et al. to accurately predict the activity of the neurons in the PPC.

This means that we can forecast the activity of neurons deep in the brain by recording simple behavioral features. The results also suggest that the PPC may be more important for setting trajectories than for processing visual images or making abstract decisions. If these findings were to be confirmed in humans, where the parietal cortex is much more complex, they might help understand better the problems that arise when this area is damaged, for example after a stroke.

DOI: https://doi.org/10.7554/eLife.42583.002

*Whitlock et al., 2012*; *Wilber et al., 2014*). Inactivating it, moreover, can impair navigation (*Save and Poucet, 2000*; *Save and Poucet, 2009*).

In principle, all of these views may be correct. Neuronal populations may benefit from a strategy of 'mixed selectivity', where neurons are tuned to mixtures of task-related variables (*Rigotti et al., 2013*). Mixed selectivity has been observed in PPC neurons of both monkeys (*Freedman and Assad, 2009*; *Park et al., 2014*; *Rishel et al., 2013*) and rats (*Raposo et al., 2014*)

To investigate how spatial navigation, body movement, and decision-making are reflected in PPC activity, we took advantage of the capabilities allowed by virtual reality (*Harvey et al., 2009*). We trained mice in a virtual navigation task that relies on visual decision-making, and recorded from populations of PPC neurons. The results closely replicated the apparent selectivity of PPC activity on choice, including the arrangement of neurons in choice-dependent activation sequences (*Harvey et al., 2012*). However, we could predict the activity of PPC neurons based on simple spatial signals: the position of the mouse in the virtual environment and its heading angle. Given the trajectories taken by the mouse, the preferences of the neurons for these attributes predicted the activation of individual neurons and explained their arrangement in sequences. Selectivity for choice, in particular, was fully explained by preferred heading. The predictions improved only slightly when we explicitly added decision as a predictor, and they worsened when we used alternative models based on vision or body movement.

## Results

We trained mice in a virtual navigation task driven by perceptual decisions (*Figure 1a–c*). Head-fixed mice performed a visual two-alternative forced-choice (2AFC) contrast-detection task by walking on an air-suspended ball (*Dombeck et al., 2010*) through a virtual corridor (*Figure 1a*, *Video 1*). One of the corridor's side walls, chosen randomly, contained a vertical grating, and mice indicated that side by turning into the left or right arms at the end of the corridor (*Figure 1b*). Successful trials were followed by a water reward, and unsuccessful ones by a brief white-noise sound. To control the task difficulty, we varied the grating's contrast randomly across trials. Accordingly, contrast exerted

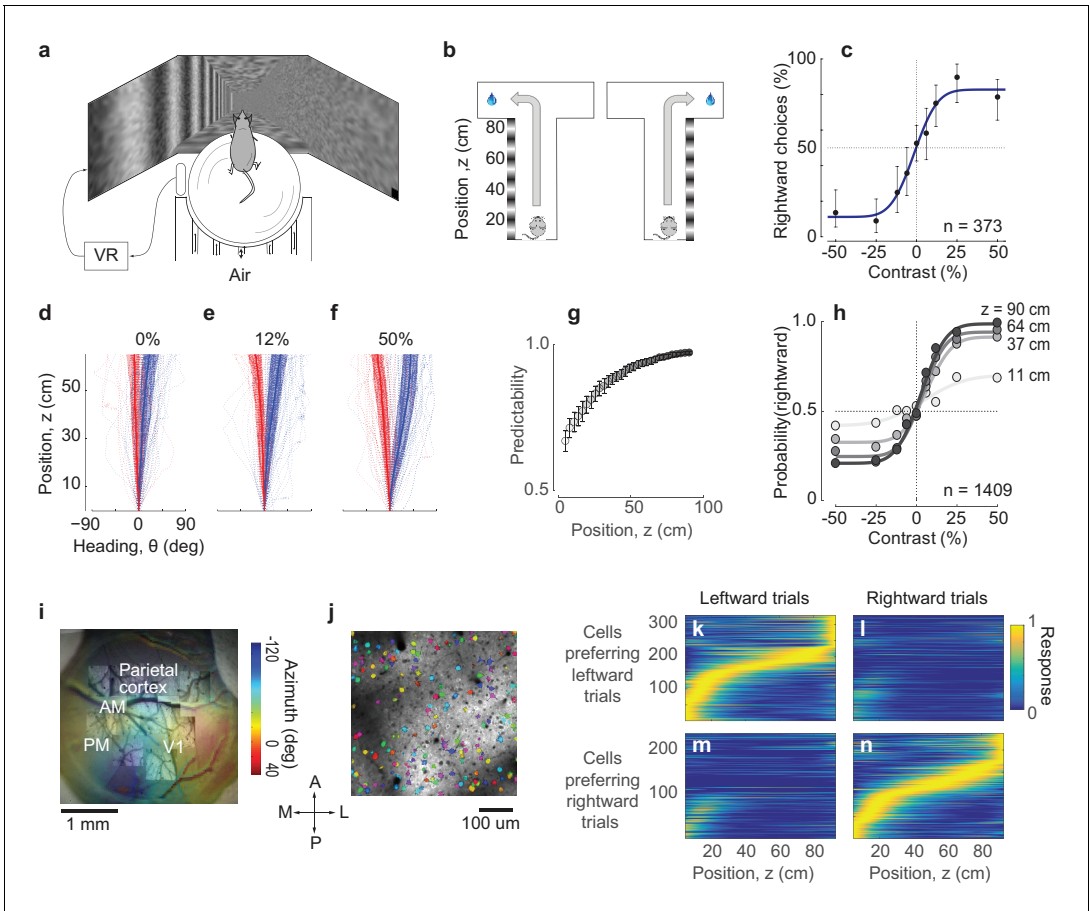

**Figure 1.** Imaging PPC activity during a navigation task driven by visual decisions. (**a**) Schematic view of the experimental setup. The monitors are positioned at 90 deg relative to each other, spanning 270 deg of the horizontal field of view. (**b**) Schematic of the virtual environment. The mouse receives a water reward for turning in the correct direction at the end of the corridor. (**c**) Psychometric curve for an example session. Negative contrasts indicate stimuli on the left wall, positive values indicate stimuli on the right wall. Error bars indicate 95% confidence intervals. (**d-f**) Examples of trajectories in position-heading coordinates within a single session (where heading was allowed to vary between −90 and 90), divided according to the whether the final choice was leftward (*red*) or rightward (*blue*). For easier trials (high contrast, (**f**), the trajectories in the corridor tended to diverge sooner than for harder trials (low or zero contrast, (**d–e**). Thick lines indicate the median θ for each z, shaded areas indicate 25th-75th percentile range of θ, dotted lines indicate individual trials. (**g**) The probability of predicting the final choice of the animal (predictability) from its heading increases as the mouse progresses through the corridor. Error bars represent s.e.m. (1409 trials in seven sessions in one mouse). (**h**) Heading provides increasingly accurate predictions of the psychometric functions as the animal progresses through the corridor (same data as in **g**), gray levels as in **g**). (**i**) Retinotopic map acquired using widefield imaging of a GCaMP6f transgenic mouse. This map was used in combination with stereotaxic coordinates to identify brain areas (*Figure 1—figure supplement 1*). (**j**) Mean fluorescence of a single plane obtained in PPC with two-photon imaging. Active cells (shown in *color*) were identified by a cell detection algorithm, and curated to include only cell bodies. (**k–n**) Choice-specific sequences of activity. PPC cells appeared to be selective to the trial outcome, some firing in trials ending with a left choice (**k**) and others in trials ending with a right choice (**n**). Also, cells were only active in a specific position in the corridor. These figures show all active cells from two example recording sessions (no cells were excluded).

DOI: https://doi.org/10.7554/eLife.42583.003

The following figure supplement is available for figure 1:

**Figure supplement 1.** Location of imaged neurons relative to somatosensory and primary visual cortices.

DOI: https://doi.org/10.7554/eLife.42583.004

a powerful influence on performance: mice frequently chose the correct side for high-contrast stimuli, and performed at chance when contrast was low or zero (*Figure 1c*).

These visual decisions strongly influenced navigation throughout the corridor, as mice typically turned towards the intended side before reaching the end (*Figure 1d–h*). To describe the navigation trajectories, we considered two variables: position along the corridor (z) and heading angle (θ). In these coordinates, the paths that ended in left and right choices progressively deviated from each

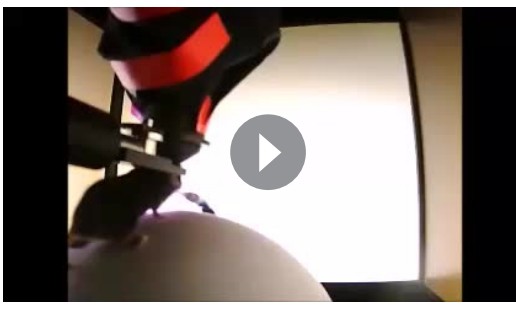

**Video 1.** A mouse performing some trials of the task.
DOI: https://doi.org/10.7554/eLife.42583.005

other: the animal started heading towards the chosen side before reaching the end of the corridor. This dependence of heading angle θ on decision was particularly clear for trials with higher contrast, which were easier (*Figure 1f*). The final choice could thus be predicted from the heading angle with increasing accuracy as the mouse reached the end of the corridor (*Figure 1g,h*).

While mice performed this task, we measured PPC population activity using 2-photon calcium imaging (*Figure 1i,j*). To identify the borders of visual cortical areas we obtained retinotopic maps using widefield imaging (*Figure 1i*), and identified PPC as a region anterior to the V1 region, along a contour of pixels that represent a retinotopic azimuth of 60–80 degrees. The average stereotaxic coordinates of this region (*Figure 1—figure supplement 1*) were close to the coordinates used in previous studies of mouse PPC (2.0 mm posterior, 1.7 mm lateral, *Goard et al., 2016*; *Harvey et al., 2012*). We then targeted 2-photon imaging to this region while the mouse was performing the task (*Figure 1j*). To obtain calcium traces from well-identified cell bodies we applied *Suite2p*, an image-processing pipeline that provides image registration, cell detection, and neuropil correction, followed by manual curation (*Pachitariu et al., 2016*).

In agreement with previous observations made in a memory-based task (*Harvey et al., 2012*), all the recorded PPC cells could be divided into two groups forming distinct, choice-dependent sequences of activation (*Figure 1k–n*). One group of cells responded primarily during trials that ended in a leftward choice (*Figure 1k,l*), and the other during trials that ended in a rightward choice (*Figure 1m,n*). Moreover, cells could be ordered so that the responses of each group of cells could be arranged in a sequence of activations (*Harvey et al., 2012*). While some cells that responded during the initial part of the task tended to fire in both trials that ended with left and right choices, the rest of the cells unambiguously fired only in one or the other of those trials.

To investigate this apparent dependence of activity on choice, we plotted the firing of individual neurons as a function of the animal's position and heading (*Figure 2a,b*). Consider an example neuron, which fired mostly during trajectories that ended with rightward choices. Plotting the neuron's activity on top of the individual trajectories reveals that the neuron tended to respond in precise combinations of position z (~60 cm into the corridor) and heading angle θ (~20 deg to the right). This combination occurred most frequently in trajectories that ended with rightward choices (*Figure 2b*) but also occasionally in trajectories that ended with leftward choices (*Figure 2a*). The neuron was active whenever a trajectory brought the mouse to the appropriate combinations of position and heading, regardless of final choice.

Indeed, a simple 'position-heading field' was sufficient to accurately predict the calcium activity of the neuron (*Figure 2c–g*). We obtained an activity map of the neuron as a function of position and heading (*Figure 2e*), and found that it could be used effectively to predict responses as a function of time (*Figure 2f*). The model performed well across trials (*Figure 2c,d*), capturing not only the overall preference for trajectories that ended in rightward choices, but also detailed differences in responses in individual trials (*Figure 2c,d*). For instance, the position-heading field correctly predicted the occasional trials when the neuron responded during leftward choices (above the black bar in *Figure 2c,d*).

Position-heading fields provided a good account of the activity across the population (*Figure 2h–j*). High correlation between data and predictions was not associated with a particular preference for position or heading: cells whose responses were accurately predicted could have a variety of position-heading fields (*Figure 2h–j*). For all these cells, the model performed well in describing trial-by-trial activity (*Figure 2—figure supplement 1*). The median correlation between model predictions and calcium traces was 27% (±16% m.a.d., n = 7,646 cells) when pooling across seven mice, and ranged from 19% to 41% in individual mice (*Figure 4—figure supplement 1*). These values are high, given that all measures were cross-validated.

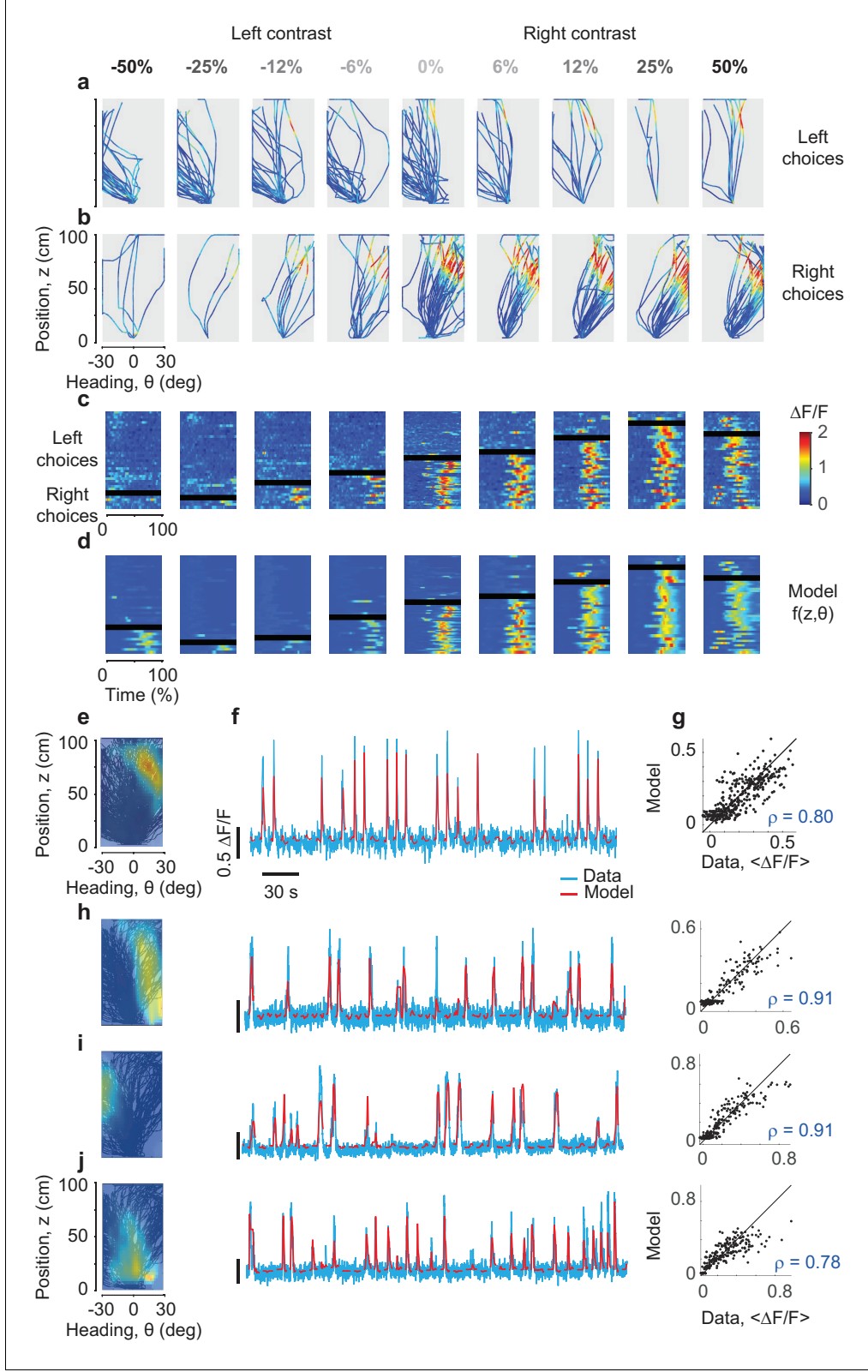

**Figure 2.** Predicting the responses of PPC neurons based on position and heading. (a,b) Activity ($\Delta F/F$) of an example neuron, plotted in pseudocolor, for trajectories that ended in leftward (a) or rightward (b) choices. Columns correspond to different stimulus contrasts and sides as indicated. Trajectories are plotted as a function of position and heading. The neuron fired (*red*) in a small region of this space, and was mostly silent (*blue*)

*Figure 2 continued on next page*

*Figure 2 continued*

elsewhere. (c) The same data, plotted as a function of normalized time. The rows in each panel correspond to trials, divided depending on whether they ended in rightward vs. leftward choices (above vs. below the *black bar*). (d) Same format as in c, but predicted by the position-heading model in e. Color scale is the same for panels a–d (*color bar* in c). (e) Position-heading field of this example neuron. Color represents the normalized $\Delta F/F$ of the neuron. (f). Model prediction (*red*) compared to the actual calcium traces (*cyan*) in representative trials. For each trial, the position-heading model was estimated without the calcium data from that trial. (g) The model provides a good explanation for the different levels of activity of the cell in different trials, with a correlation between actual data and model prediction of 0.8 (373 trials). (h–j) Examples of three other cells with position-heading fields in different locations from a different session (216 trials).

DOI: https://doi.org/10.7554/eLife.42583.006

The following figure supplement is available for figure 2:

**Figure supplement 1.** Single-trial responses and model predictions for three example cells.

DOI: https://doi.org/10.7554/eLife.42583.007

Position-heading fields were also largely sufficient to explain the arrangement of parietal cortex responses in choice-dependent sequences (*Figure 3*). For each PPC neuron we predicted responses for all trials and averaged these predictions depending on whether the trials ended in leftward choices (*Figure 3a*) or rightward choices (*Figure 3b*). The resulting predictions produce orderly,

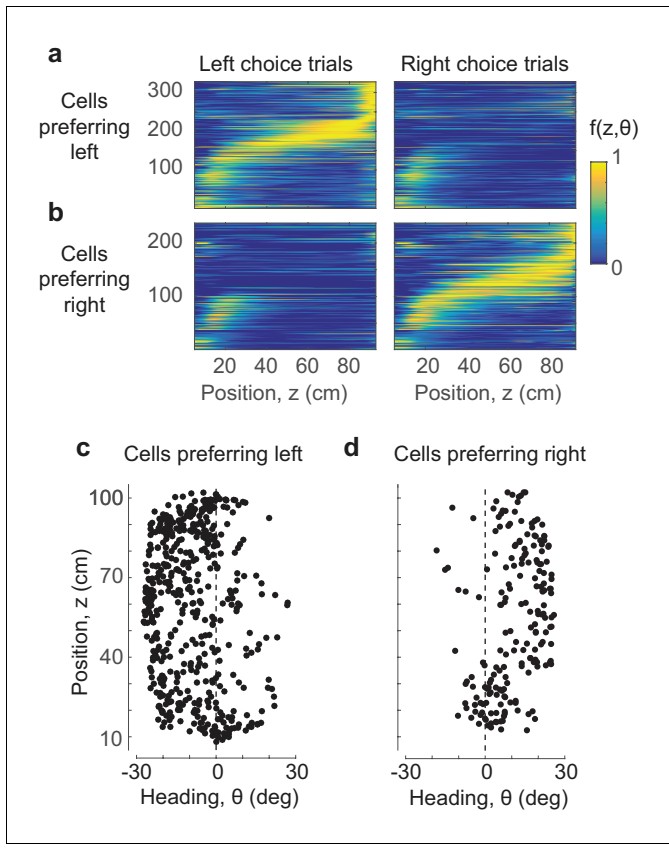

**Figure 3.** Position and heading are sufficient to explain the dependence of responses on choice. (a,b) The position-heading model correctly predicts the sequential choice-selective activations seen in the data (compare to *Figure 1k–n*, where cells are arranged in the same order). (c,d) The centers of the position-heading fields of all the cells, for cells that fired preferentially in trials that ended in leftward (c) or rightward choices (d). Almost invariably, the former preferred negative (leftward) heading angles, and the latter preferred positive (rightward) heading angles.

DOI: https://doi.org/10.7554/eLife.42583.008

choice-dependent sequences of activation, replicating the essential features of those seen in the data (*Figure 1k–n*). These choice-dependent sequences of activation emerge from a combination of two factors: the fact that mice take different trajectories in trials that end with leftward vs. rightward choices (*Figure 1d–f*), and the fact that different cells prefer different combinations of position and heading (*Figure 3c,d*).

The preferred heading of these PPC cells was thus sufficient to explain their selectivity for choice: cells with preferred leftward heading were more likely to fire when the animal headed leftward, which occurred most often when the animal ultimately chose the left arm. Indeed, cells that responded preferentially in trials ending in leftward choices almost invariably preferred negative (leftward) heading values (*Figure 3c*), and the same was true for rightward choices and positive (rightward) heading values (*Figure 3d*).

Decision was a worse predictor of PPC activity than heading (*Figure 4a–c*). A model f(z, d) where responses depend on position and decision alone is implicit in representations where neurons are arranged in choice-selective activation sequences (*Harvey et al., 2012*). It could in principle explain the sequences seen in our data (*Figure 1k–n*). However, it provided only a rough approximation of the trial-by-trial activity of individual cells, missing its graded dependence on heading angle θ (*Figure 4a*). To compare the model with the position-heading model f(z, θ), we calculated the correlation between trial-averaged measurements and model predictions, and we cross-validated the results. Before averaging activity in each trial, we excluded positions where decision and heading angle were so highly correlated as to be indistinguishable as predictors. The range of positions varied from session to session and invariably included the end of the corridor. This analysis gave a clear advantage to the position-heading model, with correlations higher by 5.6% than for the position-decision model (±9.8%, m.a.d., n = 7646 neurons, *Figure 4b*). This advantage was visible also in individual sessions (*Figure 4c*).

Furthermore, a model with all three predictors performed only slightly better than the model with position and heading alone (*Figure 4d–f*). We extended the position-heading model to obtain a model f(z, θ, d) that includes explicit knowledge of the mouse decisions (d = left or right). This extended model effectively endows the cell with two position-heading fields for trajectories that end in leftward vs. rightward choices. The extended model thus predicted slightly different curves for trials ending in left vs. right choices (*Figure 4d*). Across all neurons, the extended model performed at a similar cross-validated level as the simple position-heading model, improving the correlations by only 0.4% (±2.9%, m.a.d., n = 7646 neurons, *Figure 4e*). Similar results were seen in individual sessions (*Figure 4f*). Conversely, adding heading angle θ to a model that already includes d considerably improved the fits, with a median increase in correlation of 6.1% (±6.1% m.a.d., n = 7646 neurons). Therefore, heading angle was a much better predictor of responses than decision.

Taken together, these results suggest that much of the activity of PPC neurons in our task can be explained by two spatial attributes: position and heading. A possible caveat in this conclusion, however, is that in our experiment those spatial attributes may covary with visual and motor factors. Position and heading determined the visual scene, and the visual scene could in turn determine the activity of PPC neurons, especially given that mouse PPC overlaps at least partially with regions of higher visual cortex (*Wang and Burkhalter, 2007*; *Zhuang et al., 2017*). Likewise, position in z-θ space is itself determined by the animal's movement on the ball and therefore by motor factors such as linear and angular velocity, which may in turn determine PPC activity (*McNaughton et al., 1994*; *Nitz, 2006*; *Whitlock, 2014*; *Whitlock et al., 2012*).

To assess the role of visual factors, we ran a control experiment where the animal passively viewed a playback of visual scenes presented in the task on the same session. In this playback condition only a minority of PPC neurons maintained their preferences for position and heading, and even in those neurons the responses were much weaker (*Figure 4—figure supplement 2a*). Moreover, for the majority of PPC cells, activity in the playback condition was not predictable from the preferences for position and heading estimated during the task. This is perhaps remarkable, given that parietal areas of the mouse cortex are considered to overlap with visual areas RL, A, and AM (*Hovde et al., 2018*; *Kirkcaldie, 2012*), which contain retinotopic visual representations (*Garrett et al., 2014*; *Wang and Burkhalter, 2007*). By comparison, responses measured in primary visual cortex (V1) during the task and during playback were in better agreement (*Figure 4—figure supplement 2b*).

To assess the role of motor factors, we evaluated a model where PPC activity depends on the mouse's movement, measured by the ball's angular and linear velocities. These quantities are related

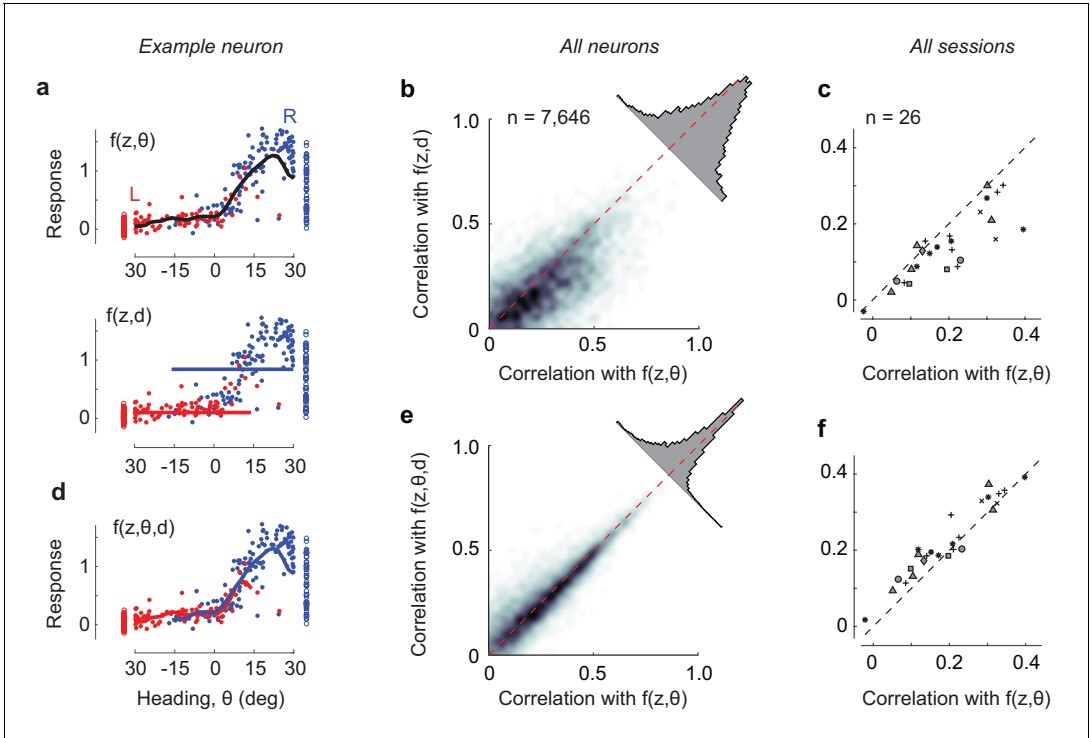

**Figure 4.** Effects of adding decision as an explicit variable. (a) Average responses of the example cell shown previously (*Figure 2a–g*) in the range of positions (z) of 60–80 cm, where the cell responds maximally, for trials ending in leftward choices (*red*) or rightward choices (*blue*). Curve: fits of the position-heading f(z, θ) model. The *bottom panel* shows the same data, fitted with a model f(z, (d) where responses depend on position z and decision, d. (b) Comparison of performance of the position-heading model (*abscissa*) and of the position-decision model (*ordinate*) for n = 7646 neurons in seven mice. Because of the vast number of neurons, data are summarized by density (*gray level*). For each neuron, model performance was measured by the correlation across trials between neuronal activity and model prediction. Neuronal activity and model predictions were trial-averaged, after excluding timepoints where decision and heading angle were highly correlated. The *histogram* shows the distribution of differences in correlation with the two models. (c): Same, but summarized as median values of correlation coefficients on a session-by-session basis. Different symbols denote different mice as indicated in *Figure 4—figure supplement 1*. (d) Same as b, for the extended model f(z, θ, d), where responses depend on position z, heading angle θ, and decision, d. The model predicts two largely overlapping curves. (e–f) Same as b–c, comparing the performance of the extended model with the position-heading model.

DOI: https://doi.org/10.7554/eLife.42583.009

The following figure supplements are available for figure 4:

**Figure supplement 1.** Quality of fits by position-heading model across all neurons in individual mice, measured by the correlation between the trial-averaged raw data and the model predictions.
DOI: https://doi.org/10.7554/eLife.42583.010
**Figure supplement 2.** Assessing the role of visual factors.
DOI: https://doi.org/10.7554/eLife.42583.011
**Figure supplement 3.** Assessing the role of motor factors.
DOI: https://doi.org/10.7554/eLife.42583.012

to the derivatives of position z and heading θ, but they are not identical because they are in mouse-centered coordinates and are unconstrained by the boundaries of the virtual corridor. The model based on motor factors performed markedly worse than the model based on position and heading in virtual reality (*Figure 4—figure supplement 3*).

We finally asked if PPC population activity and its dependence on position and heading were sufficient to decode the details of the mouse's trajectories and choices (*Figure 5*). Using established methods (*Oram et al., 1998*; *Zhang et al., 1998*) we implemented a simple Bayesian decoder (*Figure 5—figure supplement 1*). The decoder successfully predicted the position of the animal in position-heading space at individual time points (*Figure 5a*) and replicated the animal's trajectory (*Figure 5b*; *Video 2*). In predicting the final choice, in fact, the population decoder was just as good as the animal's actual heading: both showed a similar dependence on position z (*Figure 5c*) and

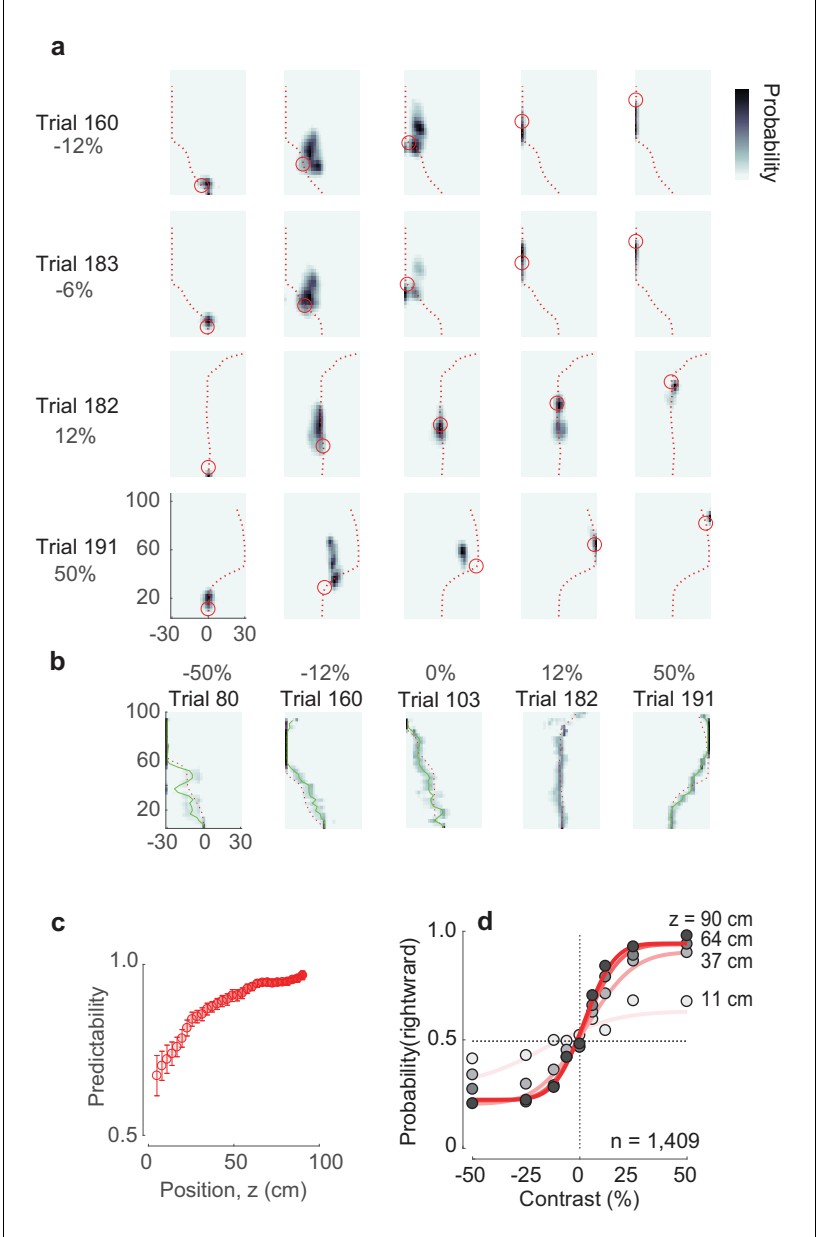

**Figure 5.** Decoding animal position and choice from neural activity. (a) The posterior estimate of the position closely follows the actual trajectory of the animal. Different rows represent different trials; different columns represent different moments in the trial. *Red* dashed line represents the trajectory of the mouse in the trial, *circle* – the actual position of the mouse in the corridor. Estimation of underlying position-heading fields $\mu_i(z, \theta)$ used for position decoding during a specific trial was performed without including the neural data of that same trial. (b) Estimated trajectories in z-θ space closely follow the actual trajectories of the mouse. The *red* dashed line represents the actual mouse's trajectory, *green* solid line represents estimated trajectory, superimposed on a *pseudocolor* representation of the underlying posterior probability distribution. (c) Choice predictability, as estimated from the decoded trajectories at different stages of the trial, from early in the trial (*faint red*) to late in the trial (*full red*). The neurometric choice predictability increases as the mouse progresses through the corridor, meaning that the final choice becomes increasingly more predictable from the neural activity. Error bars represent s.e.m. (d) Neurometric functions, estimated at different positions in the corridor (*faint red* line to *full red* line). The data points here are the same as in Figure 1h), however the curves are fit to the data points decoded from neural activity (not shown).

DOI: https://doi.org/10.7554/eLife.42583.013

The following figure supplement is available for figure 5:

*Figure 5 continued on next page*

*Figure 5 continued*

**Figure supplement 1.** Full trajectory decoding from a sequence of posterior distribution estimates.
DOI: https://doi.org/10.7554/eLife.42583.014

stimulus contrast (*Figure 5d*). This result provides further support for the view that during visually-guided navigation, populations of PPC neurons encode spatial position and heading angle.

## Discussion

We used a virtual reality task where mice use vision to decide and navigate, and we found that the activity of PPC neurons can be accurately predicted based on two simple spatial measures: position of the animal along the corridor, and heading angle. Using only these attributes, we could predict a large fraction of PPC activity during a complex task involving body movement, vision, decision, and navigation. These predictions are superior to those based purely on vision or on body movement.

These results resonate with the view that rodent PPC encodes combinations of spatial attributes (*McNaughton et al., 1994*; *Nitz, 2006*; *Nitz, 2012*; *Save and Poucet, 2000*; *Save and Poucet, 2009*; *Whitlock et al., 2012*; *Wilber et al., 2014*), but are also fully consistent with the observation that PPC cells can be divided into groups forming distinct sequences of activations depending on upcoming choice (*Harvey et al., 2012*). However, in our data, this division into choice-dependent sequences, and indeed the sequences themselves, could be explained by the effect on PPC of two measurable factors: the trajectories taken by the mouse in different trials, and the preferences of different cells for different combinations of position and heading. The selectivity of PPC cells for heading and position explained the cells' apparent selectivity for the mouse's decision.

Our results therefore appear to support a different view of PPC function to that proposed by *Harvey et al. (2012)*. Perhaps this discrepancy is due to a difference between the tasks: for example in our task, unlike the main task by Harvey et al, the spatial cues indicating the appropriate decision were visible until the end of the corridor. This might have caused the animals to employ different strategies, resulting in genuinely different types of PPC coding across the two tasks.

It is also possible, however, that the same combination of spatial factors also contributed to the results of *Harvey et al. (2012)*. Indeed, the animals' trajectories in that study did exhibit differences in heading angle that correlated with the final decision, but trials showing such differences were progressively excluded from analysis until the difference in mean heading angle no longer reached statistical significance. It would therefore be interesting to test whether preferences for position and heading may also contribute to the apparent decision-selectivity in Harvey et al.'s full data set.

In conclusion, we found that the activity of neurons in PPC during a task involving movement, vision, decision, and navigation, can be accurately predicted based on the selectivity of the neurons for two spatial variables: the position of the mouse along the corridor, and its heading angle. Consideration of this selectivity, and the mouse's trajectories through virtual space, fully accounts for the apparent formation of decision-dependent activity sequences in this task. In other words, when mice use vision to guide navigation, parietal cortex encodes

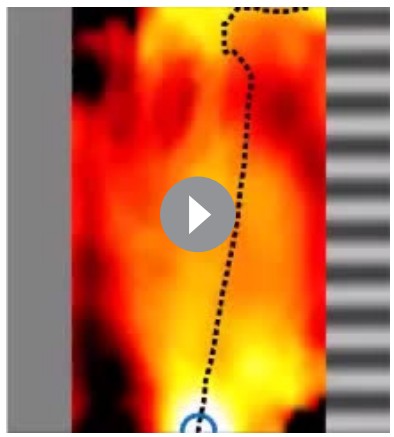

**Video 2.** Frame-by-frame decoding of mouse position from PPC population activity. The main rectangle represents the position of the mouse in the coordinates of position (z) vs. heading angle (θ). For each trial, the contrast and position of the stimulus shown in that trial is indicated by the grating shown on the left, and the trajectory of the mouse is indicated by a black dashed line. The frame-by-frame position of the mouse is indicated by a circle. The color-coded map is the log posterior distribution of the current position of the mouse estimated from population activity of PPC neurons. The peak (white) indicates the decoded position.
DOI: https://doi.org/10.7554/eLife.42583.015

navigational attributes such as position and heading rather than visual signals or abstract decisions.

Our results, however, do not completely rule out a possible role of decision signals in shaping the responses of PPC neurons. We found that adding decision as an explicit variable barely improved the fits of the position-heading model, but did not make them worse as it could have, given that the model fits were cross-validated.

Our task, in fact, cannot fully distinguish encoding of navigation variables and of decisions, because trajectory signals might strongly covary with an evolving decision variable. Such an evolving decision variable might be seen not only in covert neural signals, but also in ongoing body movements (*Dotan et al., 2018*; *Resulaj et al., 2009*; *Song and Nakayama, 2009*; *Spivey et al., 2005*). In our task, the clear dependence of trajectories on stimulus contrast is consistent with a close relationship between a decision variable and the animal's trajectories. It may not be possible to distinguish the neural encoding of the two.

Moreover, PPC activity could be influenced by a multitude of postural attributes that we did not monitor, and some of these may correlate with the apparent decision signals that improved our fits. Indeed, there is mounting evidence that postural and other motor variables act as powerful determinants of activity throughout the cortex (*Musall et al., 2018*; *Stringer et al., 2018*) including PPC (*Mimica et al., 2018*). The same observation, in fact, can be made for the apparent selectivity of PPC neurons for heading: perhaps neurons in PPC are sensitive to postural variables that in turn predict heading.

The precise motor, postural, or cognitive correlates of heading and decision need further examination, ideally with tasks explicitly designed to distinguish these correlates. In the meantime, our results indicate that the activity of large populations of PPC neurons can be explained based on simple embodied quantities such as heading and position in the environment.

## Materials and methods

We report here on experiments performed in seven mice (two C57bl/6 mice, three *Camk2a*-tTA; Ai93(TITL-GCaMP6f);*Emx1*-IRES-Cre mice, and two Ai95(RCL-GCaMP6f)-D; *Slc17a7*-IRES2-Cre-D), of both sexes, aged 80–241 days during imaging sessions. Wild-type mice were acquired from The Jackson Laboratory (www.jax.org/strain/000664). Triple transgenic mice were bred by crossing *Camk2a*-tTA (www.jax.org/strain/007004), Ai93(TITL-GCaMP6f) (www.jax.org/strain/024103), and *Emx1*-IRES-Cre (www.jax.org/strain/005628). Double transgenic were bred by crossing Ai95(RCL-GCaMP6f)-D (www.jax.org/strain/024105) with *Slc17a7*-IRES2-Cre-D (www.jax.org/strain/023527).

The recordings from the three Ai93-*Emx1* mice were made before we realized that this strain tends to show epileptiform activity (*Steinmetz et al., 2017*). We tested our recordings for this activity and the results were negative. However, we only imaged posterior regions of the cortex, where epileptiform activity can be missed (*Steinmetz et al., 2017*). For this reason, we recorded from the other strains and we ran a mouse-by-mouse analysis. This analysis did not reveal differences between the strains we used (*Figure 4—figure supplement 1*), so we pooled the data across all of them.

### Surgery

For the initial surgery the animal was anesthetized with isoflurane (Merial) at 3–5% for induction, and 0.75–1.5% subsequently. Carprofen (5 mg/kg weight, Rimadyl, Pfizer) was administered subcutaneously for systemic analgesia, and dexamethasone (0.5 mg/kg weight, Colvasone, Norbrook) was administered to prevent brain swelling. The scalp was shaved and disinfected, and a local analgesic (Lidocaine, 5% ointment, TEVA UK; or intradermal injection, 6 mg/kg, Hameln Pharmaceuticals Ltd) was applied prior to the incision. The eyes were covered with eye-protective gel (Viscotears, Alcon; or Chloramphenicol, Martindale Pharmaceuticals Ltd). The animal was positioned in a stereotaxic frame (Lidocaine ointment was applied to the ear bars), the skin covering and surrounding the area of interest was removed, and the skull was cleaned of connective tissue. A custom headplate was positioned above the area of interest and attached to the bone with Superbond C and B (Sun Medical). Then, a round craniotomy (3–4 mm diameter) was made with a fine-tipped diamond drill and/or a biopsy punch (Kai Medical). A cranial window was inserted into the craniotomy and fixed with Vetbond (3M) and Superbond C and B. The cranial window consisted of two superimposed round coverslips (WPI, #1 thickness) – one matching the inner diameter of the craniotomy (3–4 mm), and the other one providing mechanical support on the skull (typically 5 mm diameter). The two coverslips

were glued together beforehand using a Norland optical UV curing adhesive (NOA61, ThorLabs Inc.). After the surgery the animal was allowed to recover for at least three days before any behavioral or physiological measurements.

In C57Bl/6 mice, we injected the virus AAV2/1.Syn.GCaMP6f.WPRE.SV40 at a final concentration of 2.3e12 GC/ml before covering the craniotomy with the window. 100 nl of the virus was injected 300 µm below the brain surface at each of two locations targeting PPC (AP = −2 mm, ML = 1.7 mm) and V1 (AP = −3.5 mm, ML = 2.5 mm). The virus was injected at a rate of 2.3 nl every 6 s (Nanoject II, Drummond). The injection pipette was kept in place for about 10 min after the injection to allow full absorption of the virus solution in the tissue.

## Widefield imaging

To obtain maps of retinotopy we performed widefield imaging: fluorescence imaging on transgenic mice (GCaMP6f-TTA-*Emx1*-Cre), and intrinsic imaging on wildtype (C57bl/6) mice, with methods described previously (*Garrett et al., 2014*; *Pisauro et al., 2013*).

## Water control

To motivate the mice to perform the task we controlled their water intake. Mice obtained a drop of water (typically 2 or 4 µl) for every correct choice. If the water obtained during the task was below the minimum daily dose (40 ml/kg/day), mice received the rest of the dose through an appropriately weighted amount of Hydrogel. On rest days (typically Saturday and/or Sunday) the mice received all their dose through Hydrogel.

## Virtual reality setup

The mouse was head fixed with a headplate holder that did not obstruct the visual field. The mouse was free to run on an air-suspended Styrofoam ball (20 cm in diameter), whose rotation was measured by two optical computer mice (*Dombeck et al., 2010*) and then used in a custom virtual reality engine implemented in Matlab utilizing OpenGL through the Psychophysics Toolbox (*Brainard, 1997*; *Pelli, 1997*) to control the visual scene. The rotation of the ball along the axis perpendicular to the animal was responsible for forward movement in virtual reality, and the rotation of the ball along the vertical axis (at 20% gain) was responsible for turning in virtual reality. The lateral displacement of ball (rotation along the axis parallel to the animal's orientation) was ignored. The virtual reality scene was presented on three computer monitors (Iiyama ProLite E1980SD, 1280 × 1024 pixels, 60 Hz) positioned in a U-shaped configuration around the mouse spanning 270 degrees of the visual field horizontally and 75 degrees vertically. We used a multiplex video card (Matrox TripleHead2Go Digital Edition) to present the visual stimulus on three monitors in a synchronized manner. The light intensity response of the green and the blue channels of the monitors was linearized, while the red component was switched off to reduce light contamination in the fluorescence channel. In addition, to compensate for the light intensity drop-off at sharp viewing angles we attached three Fresnel lenses (f = 22 cm, BHPA220-2-6, Wuxi Bohai Optics Apparatus Electronic Co., Ltd, Wuxi, Jiangsu, China) in front of the monitors.

## Behavior

The virtual reality environment consisted of a 110 cm long T-Maze with a 20 cm wide corridor. Virtual position was never allowed to be less than 5 cm to any wall. During a trial, a vertical grating appeared on the left or right wall of the corridor. The grating was superimposed on a noise texture (20% visual contrast), which was identical on both walls and across trials. The spatial frequency of the grating was 14 cycles/m, and the resulting spatial frequency in the mouse visual field (in cycles/deg) varied depending on the distance and angle of the wall relative to the mouse. To provide additional visual flow and context, a 40% visual contrast noise pattern was applied to the floor of the corridor (and the ceiling, in some experiments). The sequence of grating contrasts was randomly drawn from a uniform distribution, with negative values indicating positions on the left wall, and positive values positions on the right wall. However, to prevent the animal from developing behavioral biases, trials ending in a wrong choice were repeated until finished correctly, at which point the next trial in the sequence was again a random trial. This 'baiting' strategy was applied mainly when the animals displayed a behavioral bias. Correct trials were indicated by a brief beep (0.1 s, 6.6 kHz) tone, while

error trials were indicated by a brief (0.2 s) white noise sound. During the inter-trial interval (~2 s) the screen was gray. Trials not finished within 45 s were timed out and a longer (3 s) white-noise sound was played. Mice typically performed between 200 and 400 trials per session (session duration 45–60 min), with typical duration of the finished trials of 6.0 ± 2.0 (median ±m.a.d.) seconds, varying between 4.7 and 11.6 s (median values) for individual animals. The behavioral session was aborted when either the animal stopped performing, or stopped consuming the water reward. Mice required different numbers of training sessions to reach behavioral performance sufficient to be considered for imaging (7 to 36 sessions, 21 on average across seven animals).

## Playback

Playback trials were run immediately after the actual behavioral session, while recording the same cells. The water spout was removed, and the visual stimulus was constructed by chopping the sequence of visual scenes into 0.5 s segments and presenting them in randomized order. To reduce the flickering effect of the visual stimulus, each 0.5 s segment was modified by sinusoidally modulating the contrast at the frequency of 2 Hz, thus smoothing the transition between sequential segments. The animal was free to run on the ball, but this had no effect on the visual stimuli presented. Typically, during these measurements the animals chose to alternate bouts of running with periods of rest.

## Two-photon imaging

Two-photon imaging (total of 33 recording sessions) was performed using a standard resonant B-Scope microscope (ThorLabs Inc.) equipped with Nikon 16x, 0.8 NA objective, and controlled by ScanImage 4.2 (*Pologruto et al., 2003*). Frame rate was set to ~30 Hz, with the field of view of ~500×500 μm (512 × 512 pixels). This frame rate was further shared between 3–5 imaging planes spanning the depth of L2/3 using a piezo focusing device (P-725.4CA PIFOC, Physik Instrumente) resulting in a 6–10 Hz effective sampling rate per cell. Laser power was depth-adjusted and synchronized with piezo position using an electro-optical modulator (M350-80LA, Conoptics Inc.). The imaging objective and the piezo device were light shielded using a custom-made metal cone, a tube, and black cloth to prevent contamination of the fluorescent signal caused by the monitors' light. Excitation light at 970 nm was delivered by an Ultra II femtosecond laser (Coherent, UK)

## Data preprocessing

Preprocessing of the two-photon data routinely included registration, segmentation, and neuropil correction (. The whole cell detection pipeline is explained in *Pachitariu et al., 2016*. ). To analyze playback experiments we also applied a deconvolution algorithm to extract spikes from the continuous calcium data (*Vogelstein et al., 2010*). To calculate ΔF/F, the baseline fluorescence $F_0$ was taken as the 20th percentile of the overall level of fluorescence of a cell.

## Estimation of z-$\theta$ maps

To estimate the position-heading field of each neuron we used a local likelihood approach (*Loader, 1999*). First, we used the data to estimate the occupancy map $M_{occ}(z, \theta)$ and the accumulated fluorescence signal map $M_{sig}(z, \theta)$. Then, we filtered both maps with a Gaussian filter, and we calculated the resulting position-heading field as:

$$F(z, \theta) = \frac{M_{sig}^{filt}(z, \theta) + \lambda \cdot F_{mean}}{M_{occ}^{filt}(z, \theta) + \lambda}$$

Where $F_{mean}$ is the mean fluorescence of the cell, and $\lambda$ is a small number used for regularization, to prevent large estimation errors in location where little or no data is available in z-θ space.

The size of the Gaussian filter $\{\sigma_z, \sigma_\theta\}$ was optimally chosen for each cell through a 10-fold cross-validation procedure. In this procedure, for each set of values $\{\sigma_z, \sigma_\theta\}$ 90% percent of the trials (training subset) were chosen to estimate $\hat{F}(z, \theta)$. Then, performance of the model $\hat{F}(z, \theta)$ was measured on the remaining 10% of the trials (test subset). On each fold the following error function was used to estimate the performance of the fit:

$$\epsilon(\sigma_z, \sigma_\theta) = \frac{\left\langle \left( \hat{F}(z(t), \theta(t)) - F(z(t), \theta(t)) \right)^2 \right\rangle_t}{\left\langle \left( F(z(t), \theta(t)) - F_{mean}^{train} \right)^2 \right\rangle_t}$$

Where $F_{mean}^{train}$ is the average $\Delta F/F$ of the training (90%) subset of the data. The procedure was repeated 10 times for different 90/10 partitions of the data. The set of filter values $\{\sigma_z, \sigma_\theta\}$, which resulted in the best overall performance of the model was chosen as optimal and used in subsequent analyses for that neuron.

### Models involving decision

Models incorporating decision were estimated using a similar cross-validation approach. The data for trials ending in leftward and rightward choices were treated separately. For each half of the data, we estimated the optimal $\sigma_z$ (for the $f(z, d)$ model) or $\{\sigma_z, \sigma_\theta\}$ (for the $f(z, \theta, d)$ model) using the same cross-validation procedure as for the $f(z, \theta)$ model, explained previously. This effectively resulted in two sub-models for each model ($f(z, d)$ or $f(z, \theta, d)$), with two sets of optimal smoothing parameters – one for trials ending in leftward choices and one for trials ending in rightward choices. The two sub-models were then taken together to predict calcium activity on the full set of trials.

### Quality of fit analysis

Depending on the model, different combinations of behavioral parameters (z-θ, z-d, or z-θ-d) were used to predict each cell's activity. For each session the data were randomly divided into 10 groups of trials (with balanced rightward and leftward trials in each group). For each group of trials, the activity was predicted using the other 90% of the data $\sigma_z$ or $\{\sigma_z, \sigma_\theta\}$. The Pearson's correlation coefficient between mean trial-by-trial actual and predicted responses was used to evaluate and compare performance of different models.

At certain positions (z) of the T-Maze (typically towards the end of the corridor), heading (θ) and final choice (d) become strongly correlated. To increase sensitivity of the comparison between the models we excluded data coming from these times (effectively, when the mouse is at certain z positions) when estimating quality of fits (*Figure 4*). For each behavioral session and for each position z we have estimated the area under the ROC (auROC) curve for two variables – heading and choice (which is a measure of how well one can be predicted from another). On a session-by-session basis, data from positions z where the auROC curve was greater than 0.95 were excluded from the estimation of fit quality.

### Analysis of playback data

Calcium dynamics measured with GCaMP6f is slow, with decay times as long as a few hundreds of milliseconds (*Chen et al., 2013*). During playback presentation of 0.5 s visual stimuli, this slowly decaying signal will cross-contaminate responses to sequential stimuli. Therefore, for the analysis of the playback experiment, where this issue is critical, we used inferred firing rate (*Vogelstein et al., 2010*) and not ΔF/F. To compare activity between the behavior and playback conditions at the corresponding 0.5 s segments, we used the correlation coefficient between the inferred firing rates (binned at 0.5 s).

### Decoding of population responses

To predict the distribution of locations in z-θ space visited by the animal during the session ($Pr(z, \theta)$) we used the position-heading fields ($\mu_i(z, \theta)$ estimated separately for each cell *i*), and employed a Bayesian decoding approach.

Below we show that in this approach, the posterior probability distribution for the animal's position in z-θ space at time *t* is:

$$\log Post(z, \theta) = -\sum_i \frac{(\mu_i(z, \theta) - r_i(t))^2}{2\sigma_i^2} + \log Pr(z, \theta) + const$$

where $r_i(t)$ is the response of cell *i* at time *t*, and $\sigma_i^2$ is the overall variance of the response of cell *i* throughout the session.

To see this, assume that the response ($\Delta F/F$, or even simply $F$) of cell $i$ at each location $(z, \theta)$ is a Gaussian random variable:

$$R_i(z, \theta) \sim N(\mu_i(z, \theta),\ \sigma_i(z, \theta))$$

where $\mu_i(z, \theta)$ is the expected fluorescence of the cell at position $(z, \theta)$, that is its position-heading field, and $\sigma_i(z, \theta)$ is the variability of fluorescence at this location. Let's make the additional assumption that $\sigma_i(z, \theta) = \sigma_i$, that is it depends on the cell $i$ but not on location in $(z, \theta)$.

Given these assumptions, the probability of this random variable to equal the value $r_i$ at time $t$ is:

$$Pr(R_i(z, \theta) = r_i(t)) = \frac{1}{\sqrt{2\pi} \cdot \sigma_i} \cdot \exp\left(-\frac{(\mu_i(z, \theta) - r_i(t))^2}{2\sigma_i^2}\right)$$

and the likelihood of a measured population response is:

$$L(z, \theta) = \prod_i Pr(R_i(z, \theta) = r_i(t)) = \prod_i \frac{1}{\sqrt{2\pi} \cdot \sigma_i} \cdot \exp\left(-\frac{(\mu_i(z, \theta) - r_i(t))^2}{2\sigma_i^2}\right)$$

Taking the logarithm yields:

$$\log L(z, \theta) = -\sum_i \frac{(\mu_i(z, \theta) - r_i(t))^2}{2\sigma_i^2} - \sum_i \log(\sigma_i) - \sum_i \log\left(\sqrt{2\pi}\right)$$

If we want to incorporate the position prior (occupancy map) to get the *posterior* probability distribution:

$$Post(z, \theta) = L(z, \theta) \cdot Pr(z, \theta)$$

$$\log Post(z, \theta) = \log L(z, \theta) + \log Pr(z, \theta) = -\sum_i \frac{(\mu_i(z, \theta) - r_i(t))^2}{2\sigma_i^2} - \sum_i \log(\sigma_i) - \sum_i \log\left(\sqrt{2\pi}\right) + \log Pr(z, \theta)$$

Dropping the constants leaves only position-dependent variables, to obtain the expression at the beginning of this section.

## Acknowledgments

We thank Charu Reddy for technical support. This work was supported by a Wellcome Trust doctoral fellowship (109004 to JJL) and by grants from the European Research Council (project CORTEX), the Wellcome Trust (095668, 095669, 205093, and 108726), and the Simons Collaboration on the Global Brain (325512). MC holds the GlaxoSmithKline/Fight for Sight Chair in Visual Neuroscience.

## Additional information

### Funding

| Funder | Grant reference number | Author |
| --- | --- | --- |
| Wellcome Trust | 109004 | Julie J Lee |
| Simons Foundation | 325512 | Kenneth D Harris<br>Matteo Carandini |
| Wellcome Trust | 095668 | Kenneth D Harris |
| Wellcome Trust | 205093 | Kenneth D Harris<br>Matteo Carandini |
| Wellcome Trust | 108726 | Kenneth D Harris<br>Matteo Carandini |
| Wellcome Trust | 095669 | Matteo Carandini |
| H2020 European Research Council | CORTEX | Matteo Carandini |

The funders had no role in study design, data collection and interpretation, or the decision to submit the work for publication.

### Author contributions
Michael Krumin, Conceptualization, Data curation, Software, Formal analysis, Investigation, Visualization, Methodology, Writing—original draft, Writing—review and editing; Julie J Lee, Data curation, Formal analysis, Funding acquisition, Validation, Writing—review and editing; Kenneth D Harris, Conceptualization, Resources, Funding acquisition, Investigation, Methodology, Writing—review and editing; Matteo Carandini, Conceptualization, Resources, Supervision, Funding acquisition, Investigation, Writing—original draft, Project administration, Writing—review and editing

### Author ORCIDs
Michael Krumin http://orcid.org/0000-0001-7356-6994
Julie J Lee http://orcid.org/0000-0002-7293-8538
Kenneth D Harris http://orcid.org/0000-0002-5930-6456
Matteo Carandini http://orcid.org/0000-0003-4880-7682

### Ethics
Animal experimentation: All experimental procedures were conducted according to the UK Animals Scientific Procedures Act (1986). Experiments were performed at University College London, under a Project Licence (70/8021) released by the Home Office following appropriate ethics review.

### Decision letter and Author response
Decision letter https://doi.org/10.7554/eLife.42583.020
Author response https://doi.org/10.7554/eLife.42583.021

## Additional files

### Supplementary files
• Transparent reporting form
DOI: https://doi.org/10.7554/eLife.42583.016

### Data availability
Behavioral and two-photon imaging data have been deposited in Dryad Digital Repository and are available at doi: 10.5061/dryad.ht3564h.

The following dataset was generated:

| Author(s) | Year | Dataset title | Dataset URL | Database and Identifier |
|---|---|---|---|---|
| Krumin M, Lee JJ, Harris KD, Carandini M | 2018 | Data from: Decision and navigation in mouse parietal cortex | https://dx.doi.org/10.5061/dryad.ht3564h | Dryad Digital Repository, 10.5061/dryad.j1fd7 |

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
