## [Decision Letter]

[Editors’ note: a previous version of this study was rejected after peer review, but the authors submitted for reconsideration. The first decision letter after peer review is shown below.]

Thank you for submitting your work entitled "Decision and navigation in mouse parietal cortex" for consideration by *eLife*. Your article has been reviewed by three peer reviewers, one of whom is a member of our Board of Reviewing Editors, and the evaluation has been overseen by a Senior Editor. The reviewers have opted to remain anonymous.

Our decision has been reached after consultation between the reviewers. Based on these discussions and the individual reviews below, we regret to inform you that your work will not be considered further for publication in *eLife*.

As you can see from the reviews below, all three reviewers found merit in your study, but they also shared a number of major concerns. These concerns included: 1) potentially epileptic mice and overall low animal numbers that would likely require substantial new experiments; 2) the ability to identify decision-related signals given the task design and analysis methods used, particularly relative to the richer navigation-centric framework; and 3) unclear comparisons to Harvey et al. and other previous results. The reviewers were confident that you would be able to carry out additional experiments effectively. However, they also agreed that the manuscript is not suitable for *eLife* in its current form.

If you are able to substantially expand the dataset and address the reviewers' concerns, then we would be willing to reconsider the work as a new submission to *eLife*. In that case, please include in the cover letter a description of the changes that you have made.

Reviewer #1:

This study uses an impressive combination of in vivo calcium imaging and navigation/choice behavior in a virtual-reality environment to try to distinguish different representations (body movement and navigation versus visual-driven decisions) in mouse parietal cortex. The data show a fairly strong and clear relationship between activation of neurons in the PPC and navigation trajectory defined by position and heading angle. These factors explain the PPC population data much better than factors related to the visual stimulus, the binary decision, and the motor patterns.

The conclusions of this study are in contrast with those reported previously that measured similar patterns of activity under very similar conditions (Harvey et al., 2012). That study focused on "choice-related activity," and noted that the activity could not be explained by the mouse's running trajectory ("The choice-specific activity could result if mice experienced different visual stimuli and running patterns on right and left trials and if PPC activity was modulated by those differences. To examine this, we first performed a multiple regression analysis to determine the potential effects of the parameters defining the mouse's running trajectory on the fluorescence changes during the delay period (Supplementary Table 1). These parameters could not explain the choice-specific activity patterns, suggesting that any differences in running trajectories between right and left trials did not trigger the activity we observed.").

In contrast, here the claim is that "we could explain these observations parsimoniously by the effect on PPC neurons of two simple navigational parameters: position in the environment and heading."

I found this study for the most part to be carefully designed, executed, and analyzed and thus represents an important step forward in our understanding of the role of this circuit in navigation and decision-related behaviors. However, I also have several major concerns:

1) My biggest concern is about how the present study distinguished navigation encoding from decision encoding. In particular, they have a reasonably sophisticated description of those two navigation parameters at all points within a trial. In contrast, they describe decision as just the binary outcome and have no model for what the evolving decision process actually looks like on a given trial. For example, is it a gradual process of evidence accumulation that might modulate activity throughout a trial? Or does it happen abruptly at random times, making it difficult to know if/when/how then neural activity might be affected? Given such issues, it doesn't seem surprising that the decision term didn't do a very good job of accounting for the neural data.

2) It seems like it would be worthwhile to directly compare the responses of units that were activated by comparable position/heading pairs but that ultimately led to different choices, or the same choice but on correct versus error trials.

Reviewer #2:

This is a somewhat intriguing study from a group of investigators who have developed a number of really nice experiments for examining neural signals in behaving animals. Although I think the key issue the authors pose-the functional role of PPC for navigation and decision making-is a really important one, this study seems to seriously miss the mark in three major ways.

1) The original Tigre (Titl; Ai93 line) mice from the Allen Institute have a gorgeous level of signal at the single-cell and wide-field levels, and are easy to use. However, it is now generally known in the field that when the Ai93 line is combined with a pan-neuronal Tet enhancer line, such as the CamKII line used here, and an excitatory Cre line, such as the Emx1 line used here, all the resulting mice are epileptic. In fact, these same investigators have published a pre-print of an article on bioRxiv (https://www.biorxiv.org/content/early/2017/05/16/138511.1) showing epileptiform activity in the cortex of this same three-line cross. There is thus a serious technical issue in the interpretation of the neural data from these animals. At the barest minimum, these results should be fully replicated in a larger set of animals using a different Tet line cross or viral expression of gCAMP6f.

2) Related to the issue of mice, I find the numbers here to be troubling. First, the authors used only 5 mice total for the study, 2 of which were wild-types and 3 of which were the Ai93 line. This is quite a low number of animals, given the first major issue above. In addition, the authors appear to be using cells as the 'n' for statistical analysis, which is a completely inappropriate approach. They should instead be using animals as the 'n', either alone or with weighting of the data from each animal. However, because the 'n' and stats for each analysis are not well described, it is almost impossible to understand from the manuscript what criteria were used.

3) Last, although the authors use the Harvey et al., 2012 paper as a foil for re-examining whether PPC neurons encode decisions, the task used in the current study is quite different. In that original paper, the cue in the virtual T maze was only presented for a short section of the animal's approach, and the animal had only a short period of time to integrate information and make a decision. In the current study, the cue is presented continuously along the approach corridor, and the animal has a much longer period of time in which to decide. The version in the current study may simply be much easier for the mouse to perform, and it is possible that either easy tasks do not require PPC circuits to be engaged or that the representation of the decision under these circumstances is relatively sparse.

*Reviewer #3:*

This paper examines the relationship of mouse PPC responses to body position and choice during a virtual navigation visual decision task. Mice navigated a virtual T-maze and were rewarded for navigating towards the side that displayed a grating stimulus on its wall. Mice were head-fixed in this virtual setup, facilitating 2-photon calcium imaging. These responses were selective for position and heading angle. Furthermore, responses were not better explained when including information about the mouse's choice. The authors conclude that PPC encodes navigational attributes rather than decisions when mice use vision to make spatial choices.

This paper is nicely written with a clear presentation of the data and analyses performed. The topic is also likely to be of interest to a wide range of researchers. My major reservation is that I am not sure that the experimental design allows for as strong a conclusion as the authors have drawn. Nonetheless, I think these findings are important, especially in light of their contrast to other high profile results reported elsewhere.

1) The authors are trying to determine the extent to which PPC encodes heading angle versus choices. However, they have trained mice in a way that heading angle embodies choice, as they show in Figure 1. Furthermore, with the visual stimulus present along the full length of the T-maze, the mouse may decide at any point before the end and/or change its mind. In this situation, if PPC were perfectly reflecting the mouse's decision process that was then embodied in its heading angle, I think that would give the same results as the authors found. So while I am reasonably convinced that PPC encodes heading in this task, I don't think the data allow the authors to exclude the possibility that it is also encoding an evolving choice.

2) Clearly, the contrast with the results of Harvey et al., 2012, are a critical component of this paper. The authors rightly consider task differences as a potential explanation of the differing results. However, I do not think the authors have done this adequate justice. Harvey et al. used a memory-guided task as opposed to a visually-guided task in the present paper. This difference is the basis of an extensive literature examining neural mechanisms of working memory. Furthermore, Harvey et al. reported this difference as key for PPC's role, having found it to be necessary when they used a memory-guided task but not a visually-guided task. The authors here describe this as follows: "…unlike Harvey et al.'s, the spatial cues indicating appropriate decision were visible until the end of the corridor. This might have caused the animals to employ different neural strategies…" In fact, Harvey et al. had already shown the task difference to cause animals to employ a different neural strategy, to use the phrasing of the present authors. Bottom line, I think this point is potentially much less subtle than portrayed.

3) The methods are not up to par for an *eLife* manuscript. The standard should be ability to reproduce the experiments. Some examples: The authors should fully describe the stimuli employed. There is incomplete description of the grating stimulus (Figure 1 seems to show a noise component to it) and no description of the competing stimuli on the other wall. The analyses require better descriptions. For instance, I found no mention of how their model incorporates choice into its predictions, an absolutely critical point for the manuscript. As a second example, there are many ways one could calculate variance explained but the reader is left guessing. For each figure panel, there should be a corresponding section of the methods that states exactly the computational steps taken to generate that panel.

[Editors’ note: what now follows is the decision letter after the authors submitted for further consideration.]

Thank you for resubmitting your work entitled "Decision and navigation in mouse parietal cortex" for further consideration at *eLife*. Your revised article has been favorably evaluated by three peer reviewers, including Joshua Gold as the Senior and Reviewing Editor.

The manuscript has been improved but there are some remaining issues that need to be addressed before acceptance, as outlined below:

In this revised manuscript, the authors have done an excellent job of responding to many of the concerns raised in the first round of reviews. In particular, they have ruled out the possibility of epilepsy in their animals, and they have substantially increased the amount of data. They also more clearly delineate key differences between this study and that of Harvey et al.

However, the reviewers expressed a few lingering concerns that we would like you to address. All of these concerns are related to points that were brought up in the first round of reviews and should not require substantial new work.

Specifically:

1) There still are concerns that the authors' interpretation of the results does not adequately address the possibility that because position and trajectory signals might covary strongly with an evolving decision variable, it might not be possible to distinguish navigation versus decision encoding. There is precedent for an evolving decision variable that can be seen in not just covert motor-planning signals in the brain, but in actual, ongoing movements, as well:

Review: Song and Nakayama, 2009; Spivey, Grosjean and Knoblich, 2005.

Put another way, it seems possible that a close relationship between navigation and the decision variable exists, which would imply that it is not possible to completely rule out that the neuronal encoding of navigation signals also at least partly reflects encoding of the ongoing decision variable – despite the strong statement that "nowhere in the paper do we claim to be studying such a thing" [i.e., an evolving decision variable].

On a related note, given the close relationship between heading and final choice, it seems likely that adding the final choice to a model with both position and heading is not expected to have much of an effect, even for a truly decision-encoding neuron. Therefore, the interpretation of "decision" (here implemented as just the final, binary choice, not an evolving decision variable) as having little effect on the neural encoding seems overly dismissive. Figure 4E, F seems to show fairly reliable effects. It might be that might be useful to provide a more nuanced discussion of how these signals might represent a combination of both navigation and decision encoding in PPC.

2) In Figure 4C and 4F, the authors plot data by sessions, but it is unclear why this is the appropriate presentation of data. Why not plot by animal to show that variance across animals does not drive the result? Or, at a minimum, color-code the sessions in 4C and 4F by animal to demonstrate that the result is not driven by mis-weighting of the data due to over-representation of a particular animal.

---

## [Author Response]

[Editors’ note: the author responses to the first round of peer review follow.]

We thank the editor and the reviewers of our previous submission (summer 2017) for their suggestions. We have now addressed all of these comments, and we believe the resulting paper is even stronger.

Specifically, we have made the following major changes:

1) We added a third strain of mice, bringing the total number of mice to 7, and we confirmed with a mouse-by-mouse analysis that the results are consistent across mice (new Figure 4—figure supplement 1).

2) Thanks to the new mice and to new analyses to more data sets in the previous mice, we substantially increased the sample from 1,922 neurons to 7,646 neurons.

3) We performed tests designed to detect epilepsy in the recordings from triple-transgenic mice at risk of epilepsy (*Emx1*-Cre;*Camk2a*-tTA;Ai93), and we found negative results (Materials and methods).

4) We introduced a more intuitive explanation of our results, including examples of trajectories where the animal first heads in a direction consistent with one choice and then makes the opposite choice (new Figure 2A, B).

5) We added an explanation that in our experiment decision does not help to predict activity over what can be predicted by position and heading (Figure 4).

6) We now explain that the strength of our study lies in its ability to predict with high precision the activity of neurons in an association area of the cortex. We can do a good job at predicting this ability, and we show that we do not need to add “decision” as a regressor.

7) We now highlight the fact that we replicate the results of Harvey et al. in terms of choice-selective sequences of activation. Ours is the first and only confirmation of such sequences by an independent laboratory since the Harvey study was published (2012).

Reviewer #1:This study uses an impressive combination of in vivo calcium imaging and navigation/choice behavior in a virtual-reality environment to try to distinguish different representations (body movement and navigation versus visual-driven decisions) in mouse parietal cortex. The data show a fairly strong and clear relationship between activation of neurons in the PPC and navigation trajectory defined by position and heading angle. These factors explain the PPC population data much better than factors related to the visual stimulus, the binary decision, and the motor patterns.The conclusions of this study are in contrast with those reported previously that measured similar patterns of activity under very similar conditions (Harvey et al., 2012). That study focused on "choice-related activity," and noted that the activity could not be explained by the mouse's running trajectory ("The choice-specific activity could result if mice experienced different visual stimuli and running patterns on right and left trials and if PPC activity was modulated by those differences. To examine this, we first performed a multiple regression analysis to determine the potential effects of the parameters defining the mouse's running trajectory on the fluorescence changes during the delay period (Supplementary Table 1). These parameters could not explain the choice-specific activity patterns, suggesting that any differences in running trajectories between right and left trials did not trigger the activity we observed.").In contrast, here the claim is that "we could explain these observations parsimoniously by the effect on PPC neurons of two simple navigational parameters: position in the environment and heading."

We see the reviewer’s desire to compare the two studies, but one must keep in mind that the tasks were different. In our paper, and especially in our revised text, we take extra care to indicate that the strength of the result is that we can predict the activity of neurons in an associative area of the cortex. We mainly replicate Harvey’s results, in a different task. Moreover, we have an explanation for the results we obtain: in our task, the neurons are selective for combinations of position and heading. This explains their apparent preference for decision.

I found this study for the most part to be carefully designed, executed, and analyzed and thus represents an important step forward in our understanding of the role of this circuit in navigation and decision-related behaviors. However, I also have several major concerns:1) My biggest concern is about how the present study distinguished navigation encoding from decision encoding. In particular, they have a reasonably sophisticated description of those two navigation parameters at all points within a trial. In contrast, they describe decision as just the binary outcome and have no model for what the evolving decision process actually looks like on a given trial. For example, is it a gradual process of evidence accumulation that might modulate activity throughout a trial? Or does it happen abruptly at random times, making it difficult to know if/when/how then neural activity might be affected? Given such issues, it doesn't seem surprising that the decision term didn't do a very good job of accounting for the neural data.

From an operational standpoint, the models where decision is a variable are models that have two functions (of position z and possibly of heading theta), one for trajectories that ended on the left and one for trajectories that ended on the right. We thus have a valid operational method to ask whether decision is a variable that needs to be taken into consideration to predict the firing of a neuron.

Keeping in mind that our task and Harvey’s task are not identical, we have used the model that Harvey et al. implicitly proposed to explain their data (where responses depend on position z and decision d) and compared it head-to-head with our model, where responses depend on position z and heading theta. This comparison appears in new Figure 4. In this comparison we have taken care to exclude data obtained in positions where decision and heading angle are so correlated that they have to have equal predictive power.

Regarding the “evolving decision process”: nowhere in the paper do we claim to be studying such a thing: we simply study what the mouse does and we relate it successfully to what thousands of PPC neurons do.

2) It seems like it would be worthwhile to directly compare the responses of units that were activated by comparable position/heading pairs but that ultimately led to different choices, or the same choice but on correct versus error trials.

We thank the reviewer for this suggestion. We have added an analysis of this kind, first for an example cell, and then for all cells. For the example cell, we show that there are clear cases of trajectories when the cell fired even though the animal ended up making the non-preferred choice; this occurred when the heading angle was the preferred one for the neuron (Figure 2A, B). As for the population, we show that a cell’s “preferred decision” can be very well predicted based on their preferred heading (Figure 3C, D). Finally, when pitching models against each other, we now analyze each session by excluding all positions z when decision d and heading angle theta were too correlated to be distinguished (new Figure 4). This approach enhances our ability to distinguish models, because in the positions where d and theta are highly correlated, it would be impossible to dissociate the two. All these analyses confirm that there is no need to add “decision” as an explanation for our data in our experiment.

Reviewer #2:This is a somewhat intriguing study from a group of investigators who have developed a number of really nice experiments for examining neural signals in behaving animals. Although I think the key issue the authors pose-the functional role of PPC for navigation and decision making-is a really important one, this study seems to seriously miss the mark in three major ways.1) The original Tigre (Titl; Ai93 line) mice from the Allen Institute have a gorgeous level of signal at the single-cell and wide-field levels, and are easy to use. However, it is now generally known in the field that when the Ai93 line is combined with a pan-neuronal Tet enhancer line, such as the CamKII line used here, and an excitatory Cre line, such as the Emx1 line used here, all the resulting mice are epileptic. In fact, these same investigators have published a pre-print of an article on bioRxiv (https://www.biorxiv.org/content/early/2017/05/16/138511.1) showing epileptiform activity in the cortex of this same three-line cross. There is thus a serious technical issue in the interpretation of the neural data from these animals. At the barest minimum, these results should be fully replicated in a larger set of animals using a different Tet line cross or viral expression of gCAMP6f.

We see the reviewer’s concern: the potential for epilepsy in some transgenic mouse strains crossed with the Emx1 line was indeed highlighted by a publication from our own laboratory (Steinmetz et al., 2017). It makes sense to ask whether this problem occurred in our mice and potentially influenced the results. We have now addressed this issue in two ways. First, we performed tests designed to detect epilepsy in the recordings from triple-transgenic mice at risk of epilepsy (*Emx1*-Cre;*Camk2a*-tTA;Ai93), and we found negative results (see Materials and methods). Second, we added data from 2 additional mice where a different transgenic strategy was used (VGLut1-IRES2-Cre-D;Ai95). The results are confirmed in these mice, as well as in the two C57 mice that were already in our study. The total number of mice is now 7, and the results are consistent across them (new Figure 4—figure supplement 1).

2) Related to the issue of mice, I find the numbers here to be troubling. First, the authors used only 5 mice total for the study, 2 of which were wild-types and 3 of which were the Ai93 line. This is quite a low number of animals, given the first major issue above.

The total number of mice is now 7, and the results are consistent across them (new Figure 4—figure supplement 1). The number of neurons is now above 7,000. These are large samples of mice and neurons.

In addition, the authors appear to be using cells as the 'n' for statistical analysis, which is a completely inappropriate approach. They should instead be using animals as the 'n', either alone or with weighting of the data from each animal. However, because the 'n' and stats for each analysis are not well described, it is almost impossible to understand from the manuscript what criteria were used.

We see the reviewer’s point and we now provide a better explanation of the analyses. We believe the reviewer is referring to the figures where we compare the different models. We now summarize the results not only by looking across neurons (throughout the main text) but also on a session-by-session basis (Figure 4C, F) and in individual mice (Figure 4—figure supplement 1).

3) Last, although the authors use the Harvey et al., 2012 paper as a foil for re-examining whether PPC neurons encode decisions, the task used in the current study is quite different. In that original paper, the cue in the virtual T maze was only presented for a short section of the animal's approach, and the animal had only a short period of time to integrate information and make a decision. In the current study, the cue is presented continuously along the approach corridor, and the animal has a much longer period of time in which to decide. The version in the current study may simply be much easier for the mouse to perform, and it is possible that either easy tasks do not require PPC circuits to be engaged or that the representation of the decision under these circumstances is relatively sparse.

We thank the reviewer for this comment. Our study resembles the study by Harvey et al. in many ways, and our results recapitulate their results. Specifically, we replicate their finding of choice-selective sequences of activation. Ours is the first and only confirmation of such sequences by an independent laboratory since their study was published (2012). This said, the two tasks are not identical, and we say this throughout the paper.

Reviewer #3:This paper examines the relationship of mouse PPC responses to body position and choice during a virtual navigation visual decision task. Mice navigated a virtual T-maze and were rewarded for navigating towards the side that displayed a grating stimulus on its wall. Mice were head-fixed in this virtual setup, facilitating 2-photon calcium imaging. These responses were selective for position and heading angle. Furthermore, responses were not better explained when including information about the mouse's choice. The authors conclude that PPC encodes navigational attributes rather than decisions when mice use vision to make spatial choices.This paper is nicely written with a clear presentation of the data and analyses performed. The topic is also likely to be of interest to a wide range of researchers. My major reservation is that I am not sure that the experimental design allows for as strong a conclusion as the authors have drawn. Nonetheless, I think these findings are important, especially in light of their contrast to other high profile results reported elsewhere.1) The authors are trying to determine the extent to which PPC encodes heading angle versus choices. However, they have trained mice in a way that heading angle embodies choice, as they show in Figure 1. Furthermore, with the visual stimulus present along the full length of the T-maze, the mouse may decide at any point before the end and/or change its mind. In this situation, if PPC were perfectly reflecting the mouse's decision process that was then embodied in its heading angle, I think that would give the same results as the authors found. So while I am reasonably convinced that PPC encodes heading in this task, I don't think the data allow the authors to exclude the possibility that it is also encoding an evolving choice.

We thank the reviewer for these insightful comments. We agree that the main conclusion of our study is the joint encoding of position and heading, and the main result is that we are able to predict the activity of neurons during navigation in an associative area of the neocortex. In addition, the data allow us to ask whether there are signals that we cannot explain simply with position and heading, which might be due to decision. We thus ask the question, and the answer we get is no. However, because we don’t know exactly when the animal makes a decision, we can only speculate that the signals we are not seeing are those associated with decision.

2) Clearly, the contrast with the results of Harvey et al., 2012, are a critical component of this paper. The authors rightly consider task differences as a potential explanation of the differing results. However, I do not think the authors have done this adequate justice. Harvey et al. used a memory-guided task as opposed to a visually-guided task in the present paper. This difference is the basis of an extensive literature examining neural mechanisms of working memory. Furthermore, Harvey et al. reported this difference as key for PPC's role, having found it to be necessary when they used a memory-guided task but not a visually-guided task. The authors here describe this as follows: "…unlike Harvey et al.'s, the spatial cues indicating appropriate decision were visible until the end of the corridor. This might have caused the animals to employ different neural strategies…" In fact, Harvey et al. had already shown the task difference to cause animals to employ a different neural strategy, to use the phrasing of the present authors. Bottom line, I think this point is potentially much less subtle than portrayed.

We see the reviewer’s point, and we have now changed the paper so its main goal is to illustrate the dependence of PPC responses on position and heading. The comparison with the results of Harvey et al. (which we are the first to replicate, 6 years after their publication) is not given as much importance in the paper.

3) The methods are not up to par for an eLife manuscript. The standard should be ability to reproduce the experiments. Some examples: The authors should fully describe the stimuli employed. There is incomplete description of the grating stimulus (Figure 1 seems to show a noise component to it) and no description of the competing stimuli on the other wall. The analyses require better descriptions. For instance, I found no mention of how their model incorporates choice into its predictions, an absolutely critical point for the manuscript. As a second example, there are many ways one could calculate variance explained but the reader is left guessing. For each figure panel, there should be a corresponding section of the methods that states exactly the computational steps taken to generate that panel.

We thank the reviewer for this pointing this out. We completely agree. We have now substantially improved the Materials and methods section to provide this information.

[Editors' note: the author responses to the re-review follow.]

The manuscript has been improved but there are some remaining issues that need to be addressed before acceptance, as outlined below:In this revised manuscript, the authors have done an excellent job of responding to many of the concerns raised in the first round of reviews. In particular, they have ruled out the possibility of epilepsy in their animals, and they have substantially increased the amount of data. They also more clearly delineate key differences between this study and that of Harvey et al.However, the reviewers expressed a few lingering concerns that we would like you to address. All of these concerns are related to points that were brought up in the first round of reviews and should not require substantial new work.Specifically:1) There still are concerns that the authors' interpretation of the results does not adequately address the possibility that because position and trajectory signals might covary strongly with an evolving decision variable, it might not be possible to distinguish navigation versus decision encoding. There is precedent for an evolving decision variable that can be seen in not just covert motor-planning signals in the brain, but in actual, ongoing movements, as well:Review: Song and Nakayama, 2009; Spivey, Grosjean and Knoblich, 2005.Put another way, it seems possible that a close relationship between navigation and the decision variable exists, which would imply that it is not possible to completely rule out that the neuronal encoding of navigation signals also at least partly reflects encoding of the ongoing decision variable – despite the strong statement that "nowhere in the paper do we claim to be studying such a thing" [i.e., an evolving decision variable].

We thank the reviewers for clarifying this point and for indicating these two very interesting papers. We had not understood it in the first round of reviews, and now we understand it. Indeed, we agree with it: it is entirely possible that by observing the physical trajectories of the mice we are observing an evolving decision process that is instantiated in their body position. We added a paragraph in Discussion where we explain this, and where we cite not only those two papers but also other two papers that seem relevant.

On a related note, given the close relationship between heading and final choice, it seems likely that adding the final choice to a model with both position and heading is not expected to have much of an effect, even for a truly decision-encoding neuron. Therefore, the interpretation of "decision" (here implemented as just the final, binary choice, not an evolving decision variable) as having little effect on the neural encoding seems overly dismissive. Figure 4E, F seems to show fairly reliable effects. It might be that might be useful to provide a more nuanced discussion of how these signals might represent a combination of both navigation and decision encoding in PPC.

We see the point. We cannot exclude that the small improvement in fit quality due to adding “decision” may be due to neural signals encoding decisions (rather than other physical attributes that we did not happen to measure). In addition to the new paragraph mentioned above, we have made some word changes in this direction.

2) In Figure 4C and 4F, the authors plot data by sessions, but it is unclear why this is the appropriate presentation of data. Why not plot by animal to show that variance across animals does not drive the result? Or, at a minimum, color-code the sessions in 4C and 4F by animal to demonstrate that the result is not driven by mis-weighting of the data due to over-representation of a particular animal.

We thank the reviewers for this suggestion. We have now changed panels C and F in Figure 4 to distinguish across mice, and we provide a legend for this in Figure 4—figure supplement 1. The results clearly show that there is no over-representation of any particular animal.